# Measuring the Degree of Academic Satisfaction: The Case of a Brazilian National Institute

**Cicero Eduardo Walter [1,2]**, **Cláudia Miranda Veloso [3]** and **Manuel Au-Yong-Oliveira [4,*]**

[1]  Federal Institute of Education, Science and Technology of Piauí, Teresina PI 64000-040, Brazil; eduardowalter@ifpi.edu.br

[2]  Department of Economics, Management, Industrial Engineering and Tourism, University of Aveiro, 3810-193 Aveiro, Portugal

[3]  GOVCOPP, ESTGA, University of Aveiro, 3810-193 Aveiro, Portugal; cmv@ua.pt

[4]  GOVCOPP, Department of Economics, Management, Industrial Engineering and Tourism, University of Aveiro, 3810-193 Aveiro, Portugal

*  Correspondence: mao@ua.pt

**Abstract:** The Brazilian National Institutes are strategic elements for the growth and development of Brazilian society since they have the purpose of meeting social and economic demands. However, for this purpose to be materialized, it is essential to develop strategies and mechanisms that consider the current educational context, marked in large part by the transformation of education into a product and the increased awareness of students who expect to have their own needs met in terms of achievement and satisfaction. Based on this premise, this research aims to present an indicator for measuring student satisfaction of students from the Federal Institute of Education, Science and Technology of Piauí-Campus Oeiras (FIEPI-Campus Oeiras), in order to provide evidence of how satisfaction has presented itself in relation to the different educational profiles present in the institution. The study was conducted with a sample of 290 students from FIEPI-Campus Oeiras. The instrument used for data collection was a questionnaire structured in two sections, in which the first was intended to obtain information to characterize the sample and the second section, composed of 14 items, aimed at measuring students' satisfaction with the institution. Descriptive, exploratory, and inferential statistical techniques were used for the data treatment and for the validation of the results. The results indicate that the students are slightly satisfied with the institution and that the average satisfaction is different in relation to the courses and technological axes.

**Keywords:** satisfaction; educational management; average satisfaction index; IFPI; Brazil

## 1. Introduction

Among the purposes and characteristics of the Federal Institutes of Education, Science and Technology (FIEPI), established by [1], is the provision of professional and technological education, at all levels and modalities, forming and qualifying citizens in view of the basic premise for professional performance in the various sectors of the economy, with an emphasis on local, regional, and national socioeconomic development as an alternative when facing social exclusion through the insertion of men and women in the labor market [1].

Accordingly, the Federal Institutes are strategic elements for the growth and development of Brazilian society in terms of two complementary and inseparable aspects. On the one hand, they meet social demands, with the formation and elevation of the population's education levels, providing better conditions of employability and the consequent insertion in the labor market. On the other hand, they meet the demand for productive capital, contributing to the process of modernization



and development of the country, by providing qualified labor that leads to substantial increases in productivity and, consequently, higher profit rates [2].

However, for the professional and technological education offered by the Federal Institutes to meet their purposes, it is essential to develop strategies and mechanisms that consider the current educational context, characterized in large part by the transformation of education into a product and by an increase in students' awareness, who expect not only a diploma but that educational institutions consider and meet their needs (at a time when having a diploma will become the norm in modern societies, thus students having to distinguish themselves by the quality of knowledge acquired, in the teaching and learning process) in terms of satisfaction and fulfillment [3,4].

Research carried out in other countries, such as the United Kingdom, Romania, Portugal, and China, has shown that academic satisfaction presents itself as a multifaceted construct and with determining characteristics particular to each country, whose emphasis has been on analyzing the influence of several variables, such as the quality of the materials used, the motivation of the teaching support team, tangible aspects of the institution, face-to-face activities, learning designs and tangentially, the influence of motivation and cognitive aspects of students on academic performance, which ultimately can be mediated by academic satisfaction [5–10].

Given this context and given the strategic importance of the Federal Institutes, the present investigation has as its main objective to present an indicator for measuring student satisfaction of students at the Federal Institute of Education, Science and Technology of Piauí-Campus Oeiras (FIEPI-Campus Oeiras), in order to provide evidence as to how it presents itself in relation to the different educational profiles present in the institution.

The justification for this is based on the fact that the understanding of student satisfaction becomes important as it is an educational management tool capable of evaluating and monitoring student satisfaction by educational institutions, further constituting an indicator that reflects how satisfaction is presented, serving as a parameter for correcting failures and implementing potential improvements that increase student satisfaction and achievement, thereby increasing their chances of staying and succeeding in institutions.

In addition to this introduction, the article is structured in six other sections. The following section is about the main concepts that guide satisfaction in general and in the student context, followed by a section on the study's objectives and research hypotheses; after this, the method used for the development of this research is described. Additionally, the results arrived at are presented, constituting the core of the present investigation, then, the results are discussed, and, last but not least, the article ends with a conclusion and the references used are set forth.

## 2. Literature Review

In general, all organizations today are concerned with satisfying their users, whether they are considered as customers or consumers, and whatever their age, although it is mostly millennials currently attending higher education [11]. Consumer satisfaction is at the core of modern marketing, in both a theoretical and practical way, based on the premise that organizations need to know the needs of their consumers in order to survive and thrive [12].

Within this context, the concept of consumer satisfaction provides an understanding of how consumers develop positive or negative affect for products, services, and brands, and how this is reflected in current buying behavior, constituting a central theoretical issue. From a practical point of view, consumer satisfaction is one of the main objectives to be achieved, since without understanding satisfaction, it is unlikely that it will be possible to establish loyalty with any brand [13].

In the literature on the topic, there is no clear and widely accepted consensus on what satisfaction is, which has limited the appropriate development of measures and the comparison of results between studies. However, consumer satisfaction can be related to an affective response that varies in intensity, with a focus on the choice of a product, its purchase, and consumption and also with a specific

time when satisfaction occurs, which can vary from one situation to another, but which is generally limited [14].

On the other hand, consumer satisfaction can also be defined according to the emphasis given to the term satisfaction, so that it can be understood as a result of an evaluation on some attribute, or as a deeper process that involves an experience of consumption in its entirety [15]. This fact is corroborated by some authors [16–19] who present satisfaction as an attitude of judgment, an affective response and/or a general assessment centered on a comparison between the expected and actual results of a given product or service.

The growth in demand, the increase of vacancies for students in higher education, and the fact that, in recent years, education, in general, has come to be understood as a product, have given rise to a greater concern with the quality of services provided by the institutions of higher education. Teaching in higher education [3,20] is now often also making use of technological tools to provide a better service [21] as in other domains in the digital era.

In addition, in the current educational scenario, students expect institutions to be concerned with satisfying their needs, using active methods for the formation of critical and complete professionals who are differentiated from others when entering the labor market [4,22], that in the specific case of Federal Institutes of Education, Science, and Technology, that have similar organizational and pedagogical structures oriented for the development of technical and technological solutions to attend social demands and regional particularities [1], become more expressive.

In this context, for institutions to attract and retain students, it is necessary to increase their satisfaction while reducing their dissatisfaction with the institution, offering educational services that, in fact, contribute to academic life, and regularly assess the students' satisfaction in order to identify strengths and weaknesses that can support the adaptation of the services provided, in order to increase student satisfaction [23,24], which in some manner has been done by Federal Institutes of Education, Science, and Technology, where there is evidence that students, in general, are satisfied with this kind of institutions [25,26].

Student satisfaction is related to students' experiences during the course, constituting an important influence factor in their permanence and in creating a positive image of the institution. Satisfaction assessment provides teachers with valuable information and feedback that encourages retrospective reflection to assist in introducing institutional changes that correct failures and increase student satisfaction and achievement [27,28].

Accordingly, satisfaction is multifaceted, largely determined by the students' perception of themselves and the environment of which they are part, including their perception of the curriculum structure and/or organization of the course, about the body of teaching staff and their involvement with students, the interpersonal relationships established between students during the course, their interest in the course, their expectations of employability and personal development, among other factors [23,29–32].

## 3. Our Objectives and Research Hypotheses

This research aims to (i) measure the academic satisfaction of students at the FIEPI-Campus Oeiras and (ii) understand how it presents itself in relation to the control variables gender, educational level, courses, and technological axes. Accordingly, the following research hypotheses were established:

**Hypothesis 1.** *FIEPI-Campus Oeiras students are satisfied with the institution;*

The framework of the research hypothesis 1 is based on [25,26] who, when conducting research on student satisfaction at Federal Institutes of Education, Science and Technology, found statistical evidence that students, in general, are satisfied with the institution, which can also be valid for the present investigation, considering that all Federal Institutes of Education, Science and Technology have similar organizational and pedagogical structures, whose purpose is the development of professional

and technological education as an educational and investigative process for the generation and adaptation of technical and technological solutions to social demand and regional peculiarities [1].

**Hypothesis 2.** *Academic satisfaction presents itself differently due to differences in educational profiles.*

**H$_{2a}$.** *Academic satisfaction presents itself differently depending on the gender variable;*

**H$_{2b}$.** *Academic satisfaction presents itself differently depending on the variable courses;*

**H$_{2c}$.** *Academic satisfaction presents itself differently depending on the educational level variable;*

**H$_{2d}$.** *Academic satisfaction presents itself differently depending on the variable technological axes.*

The conceptual framework of research hypothesis 2 is based on [16–19] for postulating that satisfaction can be considered an attitude of judgment, an affective response and/or a general assessment centered on a comparison between the expected and real results of a given product or service, which, in the case in question, is understood as the result of a comparison between an idealized image of the institution by different educational profiles, based on cognitive and affective components, with the real performance of the institution in its entirety.

## 4. Method

### Data Collection and Analysis

The instrument used for data collection was a structured questionnaire originally developed by [33], considering that the technical procedure adopted was a survey. The instrument was structured in two sections, in which the first was intended to obtain information for the characterization of the sample and the second section, composed of 14 items, had the purpose of measuring student satisfaction with the institution.

Since satisfaction is understood as a latent variable [34], the 14 items for its measurement were measured using a five-point ordinal Likert scale of Concordance, developed based on Oliver's Expectation Disconfirmation Theory (Expectation and Disconfirmation) (1980) [35]. According to this author, consumers (in the specific case, students) form expectations regarding the performance and characteristics (attributes) of a certain product or service (e.g., teachers, classes, institution structure, etc.) that are later compared to their actual performance at the time of use, leading to whether or not they generate satisfaction through confirmation or disconfirmation of the expectations generated.

To validate the data collection instrument, especially the measurement items of the latent variable, at first, the instrument was applied to a random sample of 35 students, then Cronbach's Alpha was calculated, defined as a verification measure of the proportion of variability in responses [36], having obtained a Cronbach's Alpha of 0.805, which can be considered as good reliability, being between 0.8 and 0.9. However, the analysis of the final reliability, whose application was made in the total sample of the study (290 students) presented a superior result, with Cronbach's Alpha of 0.933, showing excellent internal consistency or reliability of the data collection instrument developed.

The study was conducted with a sample of 290 students from the FIEPI-Campus Oeiras. For the treatment, analysis, and interpretation of the data, SPSS Statistics software in version 24 and Numbers in version 5.0 were used. The statistical techniques used were of a descriptive, exploratory and inferential nature to describe, analyze, and interpret the behavior of the attributes under study, especially the degree of student satisfaction and how it was presented in the study sample. For this, at first, the Average Satisfaction Indicator (IMS) was calculated, obtained by means of the simple arithmetic mean of the 14 items developed to measure satisfaction, according to the following Equation (1):

$$IMS = \frac{1}{n} \sum_{i=1}^{n} X_i \tag{1}$$

where, *n* corresponds to the number of independent variables used to measure satisfaction ($i = 1, \dots , 14$).

$$IMS = \frac{1}{14} \sum_{i=1}^{14} X_i \qquad (2)$$

where $X_1$ = Satisfaction in studying at FIEPI, $X_2$ = Expectation about the physical structure, $X_3$ = Stimulating environment, $X_4$ = Incentive of Teachers to Study, $X_5$ = Expectation about teachers, $X_6$ = Expectation about the classroom, $X_7$ = Happiness to study at FIEPI, $X_8$ = FIEPI as the right choice, $X_9$ = Expectation about the library, $X_{10}$ = Expectation about stimulating classes, $X_{11}$ = Teachers' Incentives for Curiosity, $X_{12}$ = Expectations about classes that bring real solutions, $X_{13}$ = Happiness when talking about the institution, $X_{14}$ = Expectation about re-enrolling at the institution.

The satisfaction measurement tool developed and implemented in the present research is based on the fact that in the context of Brazilian educational institutions, factors related to the teaching environment and the structural environment are the most relevant for determining academic satisfaction, as verified in other research studies [37–41].

The population of the present investigation is formed by 604 students from the FIEPI-Campus Oeiras. A sampling error of 4.15% and a significance level of 5% were assumed for the calculation of the sample size, which was determined by means of simple random sampling. In addition, a significance level of 5% was assumed throughout all of the analyses. The present research was approved on 23 January 2019, by the scientific committee of the Institutional Scholarship Program for Scientific Initiation of the Federal Institute of Education, Science and Technology of Piauí, meeting the criteria of notice no. 141 of 19 November 2018.

## 5. Analysis and Presentation of the Findings

### 5.1. Sample Characterization

Among the 290 students in the sample, 57.9% are female and 42.1% are male, with a preponderance of ages between 16–19 years (53.8%) and 20–23 years (26.7%) and average family income between R $100.00 and R $1,825.00 (82.4%). Regarding the level of education, 71.7% of the students attend high school, while 28.3% attend higher education. In relation to the variable courses and curricular years, the sample has the following distribution: Third Year of High School Integrated to the Technician in Administration (12.8%), I Module of the Subsequent Technical Course in Agriculture (11.4%), Second Year of High School Integrated to the Technician in Administration (10.7%), II Module of the Subsequent Technical Course in Informatics (9.7%), IV Full Course Module in Physics (9.0%), IV Module of the Bachelor's Degree in Administration (8.3%), II Module of the Subsequent Technical Course in Commerce (7.6%), Third Year of High School Integrated to Agriculture Technician (6.9%), VI Bachelor of Business Administration Course Module (5.9%), IV Module of the Subsequent Technical Course in Informatics (5.5%), VI Module of the Full Degree Course in Physics (5.2%), I Module of the Subsequent Technical Course in Administration (3.8%), Second Year of High School Integrated to Agriculture Technician (3.4%).

Regarding the technological axes, 49% of the total students in the sample are from the Management and Business Axis, 35.9% are from the Natural Resources Axis and 15.2% are from the Technological Axis, Information and Communication. The chosen courses were the first option on the part of 74.7% of the students, who chose Teaching Quality (25.9%) and Institution Reputation (19.3%) as the main factors that contributed at the time of their choices. In addition, 39% of the students considered themselves to be poorly informed at the time of enrolment, while 38.6% considered themselves to be well informed, pointing out that the main route of information at the time of their choices was Colleagues and Friends (32,2%) and the Institution's Advertising (21.7%). When asked whether they would recommend FIEPI-Campus Oeiras to their friends, 94.1% of the students said yes.

### 5.2. Exploratory and Inferential Analysis

The result of the Average Satisfaction Indicator (IMS), obtained by Equation (1), was 3.91 (s = 0.71), being quite close to the value 4, indicating that students are inclined towards satisfaction with the institution, in accordance with [42,43], when postulating that values between 4 and 5 on a five-point Likert scale, indicate a high level of evaluation of a given construct under analysis, responding to research hypothesis 1. Another verification can be made through the analysis of the standard deviation obtained, which was 0.71, pointing out that there was low variability around the answers about academic satisfaction by the students. The value of the IMS found is largely explained by the low values of the variables: $X_9$ = Expectation about the library ($\bar{x}$ = 3.32; s = 1.15), $X_{10}$ = Expectation about stimulating classes ($\bar{x}$ = 3.58; s = 1.03), $X_{11}$ = Teachers' Incentives for Curiosity ($\bar{x}$ = 3.76; s = 0.94) e, $X_3$= Stimulating environment ($\bar{x}$ = 3.87; s = 1.00).

To answer research hypothesis 2, it is necessary to check if there are differences in the average satisfaction for the variables gender, courses, educational level, and technological axes.

Bearing in mind that the gender variable is composed of two independent groups, the verification of the differences in the mean satisfaction was done through the application of the parametric *t*-Student test for two independent samples, whose assumptions are normal distribution or $n \geqq 30$ and unknown standard deviation. In addition, it was necessary to apply the Levene test to verify the homogeneity of variances due to the gender variable being composed of two groups of different sizes.

The proof value obtained by the Levene test was 0.960, so it can be said that the variances are not significantly different at a significance level of 5%. With regard to the *t*-Student test, whose proof value obtained was 0.395, it is possible to conclude that the average satisfaction is not significantly different in relation to the gender of the students in the sample, considering a significance level of 5%.

In order to verify the existence of differences in the average satisfaction for the courses, the One-Way ANOVA parametric test was used, which brings as basics and cumulative assumptions for its application the normal distribution in the different independent groups (verified through the Kolmogorov–Smirnov test), the homogeneity of the variances (verified through the Levene test) and the independence between the groups.

The results of the Kolmogorov–Smirnov test indicated that the normality of the variable under study could not be assumed, thus making the application of the One-Way ANOVA unfeasible. As an alternative, the Kruskal–Wallis non-parametric test was used to compare the distribution of satisfaction between different courses. The proof value obtained through the Kruskal–Wallis test was less than 0.001, indicating that it is possible to conclude that at least one of the satisfaction distributions is different for the courses analyzed at the 5% significance level.

Regarding the level of education, the verification of possible differences in average satisfaction was performed by applying the parametric *t*-Student test for two independent samples, first using the Levene test to verify homogeneity of the variances, considering the differences in the sizes of the two groups of the sample.

The proof value obtained by applying the Levene test was 0.583, so it is possible to conclude that the variances are not significantly different between the groups, assuming a significance level of 5%. In addition, the proof value obtained through the application of the *t*-Student test was 0.371, allowing to affirm that there are no significant differences in the average satisfaction for the educational levels of the sample, taking into account a significance level of 5%.

For the technological axes, the verification of differences in satisfaction is possible through the application of the One-Way ANOVA parametric test, which, as previously mentioned, has a series of basics and cumulative assumptions for application, such as the normal distribution in the different independent groups, homogeneity of variances, and independence between groups.

With the aid of the Kolmogorov–Smirnov test, it was possible to verify that the variable does not follow a normal distribution, requiring once again the alternative application of the Kruskal–Wallis non-parametric test to compare the distributions of satisfaction for the different technological axes. The proof value obtained by applying the Kruskal–Wallis test was 0.004, so it is possible to conclude that

at least one of the distributions of satisfaction is different for the various technological axes analyzed, assuming a significance level of 5%.

## 6. Discussion

Based on what was previously presented, it can be concluded that research hypothesis 1 was fully validated, since the IMS value obtained was 3.91 (0.71), allowing us to infer that students are slightly satisfied with the institution. The results that confirm the research hypothesis 1 are in accordance with the results of other authors [20,25,26] who, despite finding statistical evidence that students are slightly satisfied with the institution, point out that the perception of the lack of infrastructure resources (physical environment, e.g., library and laboratories) and gaps in the teaching infrastructure (faculty, e.g., stimulating and challenging classes) have a negative impact on student satisfaction, which is also true for the present investigation.

On the other hand, research hypothesis 2 has been partially validated, since satisfaction is different for students when considering courses and technological axes. The justification for this is based on the fact that satisfaction is multifaceted, largely determined by the perception that students have of themselves and the environment of which they are part, including here their perception of the curricular structure and/or organization of the course, about the teaching staff and their involvement with students, the interpersonal relationships established between students during the course, their interest in the course, their expectations of employability and personal development, among other factors which vary considerably in relation to the institution's courses and technological axes [23,29–32].

Without a doubt, our results point towards the existence of more demanding students (regarding the physical environment as well as the human factor, namely, faculty members' abilities to teach and engage students). This is in line with the proliferation of degrees and academic courses now available, meaning that performing well and making a good salary in the workplace is thus that much more difficult. As a consequence, we see that the measuring of student academic satisfaction is a growing concern, being increasingly more important for higher education institutions such as the Federal Institute of Education, Science and Technology of Piauí-Campus Oeiras (FIEPI-Campus Oeiras). One is only as good as what one measures and, thus, planning also takes on a new and more important role: One must plan and establish a gap (which needs to be closed) between the "as is" situation (identified by a quality assurance system) and the "to be" desired situation for the near future (the result of benchmarking and strategic priorities given limited resources).

Hence, gone are the times when students would be satisfied with their diploma in itself [44,45], requiring instead practical and usable lectures [46]. Nowadays, students are very aware of the soft and hard skills (e.g., teamwork and leadership abilities vs. specific measurable knowledge, such as in computer programming or accountancy) in high demand in the marketplace and even students graduating with high marks may be extremely dissatisfied if they are unable to land a [good] job. The end result may thus seem, to a certain extent, unattainable as student perceptions of what they require an academic degree to give them have become much harder to meet and satisfy. Students want to be entertained (by interesting lectures), they want to learn (preferably for the long-term rather than the short-term), and they want the latest technological resources [21] to be available to them while studying (including access to numerous databases such as Scopus and ISI Web of Science, as well as free access to software suites such as the statistics package IBM SPSS). All of the above will contribute to an institution's reputation, which in turn, will "rub off" on the students who graduate from the institution being, as a consequence, recognized by employers in diverse industries. We are in a new age of education [11] and institutions need to react accordingly.

## 7. Conclusions

As previously mentioned, the main objective of this investigation was to present an indicator for measuring student satisfaction of students at the Federal Institute of Education, Science and Technology

of Piauí-Campus Oeiras, in order to provide evidence of how it presents itself in relation to the different educational profiles present in the institution, establishing the following research hypotheses:

**Hypothesis 1.** *FIEPI-Campus Oeiras students are satisfied with the institution;*

**Hypothesis 2.** *Academic satisfaction presents itself differently due to differences in educational profiles.*

In general, the students in the present investigation are satisfied with the institution, since the value obtained from the IMS was 3.91 (0.71), leaving the investigation hypothesis 1 fully validated. However, the research hypothesis 2 was only partially validated, considering that satisfaction is diverse only due to the variable courses and technological axes.

The results obtained in the present investigation provide an important practical contribution to the monitoring and management of student satisfaction by educational institutions since it presents an indicator that reflects how satisfaction is presented, serving as a parameter for fault corrections and the implementation of potential improvements that increase the satisfaction and achievement of students, thereby increasing their chances of staying and their success in the institution. As theoretical contributions, we emphasize the fact that the results obtained in the present investigation confirm the findings of other authors [20,25,26] who specifically studied the Federal Education Network, making it clear that although students are satisfied with the institution, factors such as access to adequate infrastructure, stimulating faculty, and educational environment are of paramount importance for increasing student satisfaction. In addition, the results found in the present investigation are in line with other investigations carried out in other countries, such as the United Kingdom, Romania, and Portugal [7–10]. Specifically, the results obtained confirm that elements that make up the learning design instituted by the institutions, which ultimately determine the quality of academic services in a tangible as well as an intangible way, exemplified in the present investigation by the expectations that students have about classes, about support services, such as access to the library, and the educational environment as a whole, have a significant influence on student satisfaction, as pointed out by [8,10]. In a complementary way, the results of the present investigation bring evidence that the expectations that students develop about teachers, as well as their interactions with them, have a significant impact on their satisfaction levels. In this sense, according to investigations carried out by [7,9] in Portugal and Romania, the motivation and satisfaction of teachers become crucial for academic satisfaction to be achieved.

As limitations, it is worth mentioning that the investigation was carried out in only one of the institution's campuses, making it impossible to transfer the results to other campuses and other institutions of the Federal Education Network (External Transferability), considering that each one has particularities linked to its context, geographic location, teaching modalities, among others.

Refs [47,48] found evidence that a good relationship between students, colleagues, and teachers is essential to reduce student dropout, in addition to the fact that life satisfaction variables, as they are characterized as a global assessment, and affective relationships are related.

Accordingly, for future studies, it is suggested to investigate the relationship between academic satisfaction and student dropout and the influence that academic satisfaction has on life satisfaction.

**Author Contributions:** Conceptualization, C.E.W.; formal analysis, C.E.W.; investigation, C.E.W. and M.A.-Y.-O.; methodology, C.E.W.; C.M.V. and M.A.-Y.-O.; resources, C.M.V. and M.A.-Y.-O.; supervision, M.A.-Y.-O. and C.M.V. All authors have read and agreed to the published version of the manuscript.

**Funding:** Personal Funding.

**Conflicts of Interest:** The authors declare no conflict of interest.

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
