# Peer review of "Measuring the Degree of Academic Satisfaction: The Case of a Brazilian National Institute"

_education, doi:10.3390/educsci10100266_

Round 1

Reviewer 1 Report

Thank you for inviting me to review the study " Measuring the Degree of Academic Satisfaction: The case of a Brazilian National Institute". The authors present an indicator for measuring student satisfaction of students, in order to provide evidence of how satisfaction has presented itself in relation to the different educational profiles present in the Brazilian National Institute. Although this study could have scientific interest, some important aspects should be reviewed by the authors. I hope that my opinions will help shape your research article suitable for review. The followings are my comments:

  • Format: The sections are not appropriate. The authors must follow the instructions for authors: https://www.mdpi.com/journal/education/instructions

It is not possible to review this manuscript in the current form.

  • With respect to the research ethics, approval from an ethics committee should have been obtained before undertaking the research. At a minimum, a statement including the date of approval and name of the ethics committee or institutional review board should be cited in the “Materials and Methods” section of the article.

Author Response

 Reviewer 1: On the approval by an ethics committee of the institution.

The response for this question is on the last three lines of Research Methodology.

“The present research was approved on January 23, 2019, by the scientific committee of the Institutional Scholarship Program for Scientific Initiation of the Federal Institute of Education, Science and Technology of Piauí, meeting the criteria of notice no. 141 of November 19, 2018.”

Reviewer 2 Report

Thank you very much for the opportunity to review this manuscript, which describes the evaluation of academic satisfaction from one of the Federal Institutes of Education, Science and Technology in Brazil.

The aim of the work and the research hypotheses were precisely defined. The key question that arises after reading the manuscript is "Why did the authors decide to develop this particular satisfaction assessment tool?" In previously published studies the variety of approaches was presented. It seems crucial to justify in the Introduction section, why this, and no other tool was used and what factors specific to the Brazilian context (similarities and differences in regard to other educational systems) should be taken into account when interpreting presented results by readers from other countries.

When submitting a work to an international journal, such as the Education Sciences, the authors should pay more attention to the issue of student satisfaction researches conducted in countries other than Brazil. Both Introduction and Discussion sections lack such wider view supported by references published in English. The tool authors referred to (line 141) was published in a journal which is not included in the Journal Citation Reports, and in a language that made it impossible for the reviewer to verify the type of validation it underwent.

It should be also emphasized that in the literature, student satisfaction may be analysed twofold. In journals orientated toward marketing issues, education may indeed be "understood as a product" (Line 88) and student satisfaction may be synonymous with customer satisfaction. However, in journals published in the area of pedagogy and andragogy (focused more on teaching and learning theories), students' satisfaction is thought to be an important element of building learners engagement, resulting in increased or decreased effectiveness of the educational process.

Finally, the framework for this study was the Expectation Disconfirmation Theory, common for marketing literature.

Considering the above, it is suggested that the manuscript should be (1) submitted to a local journal, or (2) supplemented with the above-mentioned issues, valuable from the point of view of readers from other countries, and sent to another marketing-orientated journal.

Author Response

Reviewer 2: About establishing the context with other countries.

The response for this question is on the fourth paragraph of Introduction.

“Research carried out in other countries, such as the United Kingdom, Romania, Portugal, and China have shown that academic satisfaction presents itself as a multifaceted construct and with determining characteristics particular to each country, whose emphasis has been on analyzing the influence of several variables such as the quality of the materials used, the motivation of the teaching support team, tangible aspects of the institution, face-to-face activities, learning designs and tangentially, the influence of motivation and cognitive aspects of students on academic performance, which ultimately can be mediated by academic satisfaction [5] [6] [7] [8] [9] [10]. “

Reviewer 2: Why was this specific tool of measurement used and not another one?

The response for this question is on before the last paragraph of Research Methodology.

“The satisfaction measurement tool developed and implemented in the present research is based on the fact that in the context of Brazilian educational institutions, factors related to the teaching environment and the structural environment are the most relevant for determining academic satisfaction, as verified in other research studies [37] [38] [39] [40] [41].”

Editor: About placing a justification of the research hypotheses/theoretical support.

The justifications for the research hypothesis 1 are on the second and third paragraphs of the section “2.2 Satisfaction in the Academic Environment”.

“In addition, in the current educational scenario, students expect institutions to be concerned with satisfying their needs, using active methods for the formation of critical and complete professionals who are differentiated from others when entering the labour market [4] [22], that in the specific case of Federal Institutes of Education, Science, and Technology, that have similar organizational and pedagogical structures oriented for the development of technical and technological solutions to attend social demands and regional particularities [1], become more expressive.

In this context, for institutions to attract and retain students, it is necessary to increase their satisfaction while reducing their dissatisfaction with the institution, offering educational services that in fact contribute to academic life, and regularly assess the students’ satisfaction in order to identify strengths and weaknesses that can support the adaptation of the services provided, in order to increase student satisfaction [23] [24], which in some manner has been done by Federal Institutes of Education, Science, and Technology, where there is evidence that students, in general, are satisfied with this kind of institutions [25] [26]. “

The justifications for the research hypothesis 2 are on the last paragraph of the section “2.1 The Satisfaction Concept”.

“This fact is corroborated by some authors [16] [17] [18] [19] who present satisfaction as an attitude of judgement, an affective response and/or a general assessment centred on a comparison between the expected and actual results of a given product or service.”

Round 2

Reviewer 1 Report

Thank you for inviting me to review the review version of the study " Measuring the Degree of Academic Satisfaction: The case of a Brazilian National Institute". The authors present an indicator for measuring student satisfaction of students, in order to provide evidence of how satisfaction has presented itself in relation to the different educational profiles present in the Brazilian National Institute. Although this study could have scientific interest, some important aspects have not been reviewed by the author:

  • Format: The sections are not appropriate. The authors must follow the instructions for authors: https://www.mdpi.com/journal/education/instructions

It is not possible to accept this manuscript in the current form.

Author Response

The sections have been re-arranged more in accordance with classic research structures. 

New text is included in green:

- The final part of the introduction has been improved.

- Section 5.2 has been completely re-written.

- The conclusion also has new text in green (we now go deeper).

Reviewer 2 Report

Thank you for providing an improved manuscript.

Taking into account the iThenticate report, Section 4.2. should be rephrased to avoid plagiarism and references should be added properly.

I would also suggest to clearly indicate similarities and differences between obtained results and those obtained in studies previously conducted in other countries. Although Authors supplemented the text with reports from other countries, the comparison seems too superficial.

Author Response

Thank-you for your very useful comments.

The sections have been re-arranged more in accordance with classic research structures. 

New text is included in green:

- The final part of the introduction has been improved.

- Section 5.2 has been completely re-written.

- The conclusion also has new text in green (we now go deeper). 

Round 3

Reviewer 1 Report

The authors have responded satisfactorily to my comments.

Author Response

Thank you